# Rice-Based Gluten-Free Foods and Technologies: A Review

**DOI:** 10.3390/foods12224110

**Published:** 2023-11-13

**Authors:** Jiyoung Park, Hong-Sik Kim

**Affiliations:** Department of Central Area Crop Science, National Institute of Crop Science (NICS), Rural Development Administration (RDA), 126 Suin-ro, Kwonseon-gu, Suwon 16429, Gyeonggi, Republic of Korea

**Keywords:** gluten free, rice product, food processing, food technology

## Abstract

Rice, one of the most widely consumed staples worldwide, serves as a versatile gluten-free substitute. However, review articles on technological developments in grain-free production focusing on rice are scarce. This review assesses various research results concerning the quality attributes of rice-based gluten-free foods, including bread, pasta, and beer. To optimize the key attributes in processed products, such as dough leavening in bread and the physical and cooking properties of noodles and pasta, research has focused on blending different gluten-free grains and incorporating additives that mimic the gluten function. Additionally, various processing technologies, such as starch preprocessing and extrusion puffing processes, have been employed to boost the quality of rice-based gluten-free products. Today, a variety of products, including bread, noodles, and beer, use rice as a partial replacement for barley or wheat. With rapid advancements in technology, a noticeable portion of consumers now shows a preference for products containing rice as a substitute. This trend indicates that rice-based gluten-free foods can be enhanced by leveraging the latest developments in gluten-free product technologies, particularly in countries where rice is a staple or is predominantly cultivated.

## 1. Introduction

Gluten is a protein composed of a combination of gliadin (prolamin) and glutenin, primarily found in wheat and characterized by the formation of a unique structure during food processing [1]. The proper mixing and hydration of flour result in gluten forming a three-dimensional protein network that is essential for creating a viscoelastic dough matrix, particularly in processes such as baking. In addition to playing a structural role, gluten possesses water-binding and viscosity-generating properties, which are preferred in food preparation, making it a popular food additive [2]. While many grains contain gluten, wheat, barley, and rye are the most common sources, each with its specific prolamin—gliadin, hordein, and secalin, respectively [3]. Oats contain a prolamin known as avenin [4]. Moreover, concerns about gluten content arise because oats may be mixed with rye or barley during harvesting, transport, storage, and processing [5]. However, the consumption of gluten and specific prolamins can lead to various gluten-related disorders [6]. Among these, celiac disease is a chronic condition affecting approximately 1% of the US population (about 3 million people); non-celiac gluten sensitivity is reportedly more common, affecting around 6% of the population [7]. Celiac disease triggers an autoimmune response, leading to inflammation of the small intestinal villi [7]. Although gluten intolerance primarily affects the small intestine, it often manifests as systemic symptoms, including diarrhea, anemia, weight loss, muscle spasms, chronic fatigue, and bone pain [8]. Strict adherence to a gluten-free (GF) diet is the sole treatment for these gluten-related disorders.

Today, not only those with gluten-related disorders but also many consumers without such conditions avoid or limit gluten for dietary regimes or expected health benefits. Consequently, the global market for GF products is soaring, with consumption predicted to increase from USD 6.7 billion to USD 14 billion between 2022 and 2032 [9]. Regulations mandate that GF foods, unless reduced to below 20 ppm via specific processing, cannot be made using barley, wheat, rye, or their hybrids. The “GF” code on US food labels reflects this, indicating that the unavoidable gluten content in food should not surpass 20 ppm [10]. A GF diet comprises natural GF foods, such as GF grains, pseudocereals, fruits, vegetables, and meat, and specially manufactured GF products, where GF flour substitutes for wheat flour [11]. GF grains include corn, rice, sorghum, and millet, and pseudocereals include buckwheat, quinoa, and amaranth [12]. Kaur et al. reported that several studies have focused on the importance of processing GF grains and the effect on the quality of their GF foods. Chickpea, lentils, and brown beans are also used as raw materials for GF products. Most of the raw materials have higher protein or dietary fiber contents than rice, which are also higher in GF cakes. However, to improve the physical properties and quality of GF products, it is important to select a suitable addition ratio [13].

Numerous review articles have discussed various studies related to technological developments in GF production. Some notable examples include technological and nutritional challenges in the GF diet [14], technology for GF bread and bakery product development [15], approaches for improving the quality of GF pasta and bread [16], advanced properties of GF cookies, cakes, and crackers [5], substitutes for gluten in GF beer, bread, and pasta [17,18,19], and natural hydrocolloids improving GF bread and cake properties [20]. However, the use of such food additives not only leads to excessive spending on ingredients but also potentially poses risks due to certain allergens found in enzymes or proteins [21]. The current consumer preference for high-quality GF products, alongside a desire for clean-label certification highlighting the use of natural, additive-free materials, signals the need for a fresh approach for good GF production [22]. Reports have also discussed various applications of GF grains [23]; however, rice, a global staple grain, has not been included in these reviews.

This review assesses recently published research results concerning the quality attributes of rice-based GF foods, including bread, pasta, and beer. It also aims to set the future research direction for enhancing rice-based GF foods by leveraging the latest GF product development technology trends in countries where rice is a staple or is predominantly cultivated.

## 2. Rice as a GF Material

Rice, following corn and wheat, is among the most extensively consumed grains worldwide. It serves as a staple food for over one-third of the global population and is cultivated on every continent [24]. Contributing to approximately 20% of the global human calorie intake, rice holds considerable importance in human nutrition [25]. Owing to its lack of gluten proteins, rice is a preferred choice for GF food materials and is appreciated for its mildness, hypoallergenic properties, and easy digestibility. Additionally, it is low in fat, sodium, protein, and fiber [26].

Rice is classified into brown and white rice depending on the degree of milling and can be utilized in either form according to nutritional needs. Figure 1 shows the chemical composition of rice as a gluten substitute and the overall schematic of GF food development technology using rice. White rice mainly consists of starch (78%), supplemented by proteins (6.3–7.1%), crude ash (0.3–0.8%), crude fat (0.3–0.5%), and crude fibers (0.2–0.5%). Brown rice, containing bran, which is rich in nutrients other than starch, such as proteins (11–15%) and fat (15–20%), is composed of starch (66%), proteins (7.1–8.3%), crude fat (1.6–2.8%), crude fibers (0.6–10%), and crude ash (1.0–1.5%) [27].

Rice flour is advocated as an ingredient for GF products. However, rice bread, lacking a gluten network, has demonstrated significantly lower quality compared to wheat bread due to its reduced viscoelasticity [28]. In the production of rice noodles, the absence of gluten in rice flour creates challenges in forming a cohesive dough structure, leading to technical and qualitative issues [29]. Numerous research endeavors have been made in the domain of rice-based GF food development. A range of additives, such as enzymes, emulsifiers, hydrocolloids, proteins, gelling agents, and glutathione, have been employed for producing GF rice bread [21,30,31,32]. The production of high-quality GF pasta is feasible by employing accurate amounts of protein, moisture, and hydrocolloids and selecting suitable cooking techniques and formulations to substitute the gluten network [33]. Concurrently, rice with high amylose and less than 5% damaged starch content has been identified as a crucial element in augmenting the specific volume of GF rice bread, eliminating the need for other additives [30,34,35].

For the development of rice-based GF products, it is crucial not only to enhance the physical properties with additives or processing technology, such as physical treatment, but also to cultivate appropriate rice cultivars. Additionally, in line with the high consumer preference for natural products, there is a substantial interest in natural rice materials with advanced functionality or excellent processing properties. Such consumer preferences and market demands have influenced the development of diverse rice varieties in Korea. Rice consumption in Korea has consistently decreased over the past 30 years due to a shift from home meals to dining out, increased consumption of wheat flour foods, such as bread and noodles, alterations in dietary habits, and changes in government policies [36]. However, the demand for rice product development has surged for of a multitude of reasons, such as heightened interest in health foods due to the increase in obesity, a growing preference for GF foods, production surplus, and the need to substitute imported wheat owing to the escalation in international wheat prices. Consequently, there has been a surge in interest in rice apt for processing or rice with health functionality. The damaged starch content in rice, resulting from dry milling, differs with the rice variety. Generally, rice is wet milled during processing, leading to a reduction in the damaged starch content [37]. However, this process is laborious and expensive, as it uses an extensive amount of water and involves drying [38]. Therefore, the demand for a rice variety suitable for dry milling, similar to wheat flour, has also increased. In response to this demand, the National Institute of Crop Science in Korea has recently developed two rice varieties: Dodamssal, containing resistant starch, and Baromi 2, with a floury endosperm [39,40]. These rice varieties, unlike the transparent-looking regular rice filled with polygonal starch particles, do not soak in water and shatter easily, as the starch particles are round and not densely arranged. Moreover, they resemble glutinous rice in their milky-white appearance [39]. Nevertheless, such materials should be assessed alongside processing technology as they are rice, not gluten-containing wheat. The unique rice materials and milling methods utilized in GF product research are addressed in the following section on rice-based GF products.

**Figure 1 foods-12-04110-f001:**
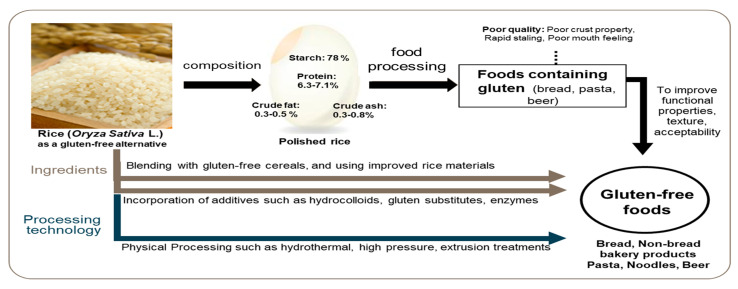
Chemical composition (%) of milled rice grain, and processing technologies for rice-based gluten-free foods [27,41].

## 3. Rice-Based GF Food Products

### 3.1. Rice-Based GF Bread and Bakery Products

Traditional gluten-containing food items, such as bakery products, now offer a range of GF alternatives that use GF cereals and pseudocereals, such as rice, corn, quinoa, millet, and amaranth, as their base ingredients [42]. Among these, rice requires the addition of technological and nutritional functional components, including proteins from various sources or hydrocolloids, to attain an adequate bread volume, crumb softness, and shelf life during the process of bread making. This addition compensates for its lower protein content and lack of gluten compared to wheat [15]. Table 1 presents the research results for quality improvement related to bread and bakery products using rice as the main ingredient. Various rice varieties have been intensively tested for the development of rice-based GF bread. When using commercially available rice or standard rice flour as ingredients, the favorable conditions for producing rice bread with a higher specific volume are a lower damaged starch content (within 4–6%), coarser particles (132–200 μm), the use of ordinary rice (as opposed to glutinous rice) owing to the difference in the amylose content, and the use of short-grain (Japonica type) rice. Additionally, cakes made with high-protein (approximately 12%) brown rice demonstrated a superior specific volume and softness compared to those made with other brown rice types or standard white rice [22]. Considering that these processing characteristics of bread and bakery products are based on the properties of the raw material (rice), it can be assumed that not only the unique characteristics of rice varieties and their nutritional components but also the milling methods that affect the particle size and damaged starch content are important. Thus, for enhancing the quality of rice bread and bakery items, thoughtful selection of the appropriate rice material is crucial.

When rice flour is treated with thermoase, a commercial protease, the quality of rice bread improves, with an increased loaf volume and decreased crumb hardness. Pretreating rice flour with gelatinization at 80 °C for 2 min increases the loaf volume and porosity [43,44]. Sourdough, a naturally mixed comprehensive strain starter used worldwide since ancient times, is traditionally made by mixing grain flour with water and fermenting the dough with natural lactic acid bacteria and yeast from wheat flour or the surrounding environment [45]. This fermentation process produces various metabolic products, including amino acids and short-chain fatty acids, that can enhance food quality and flavor. The addition of 30% sourdough to rice dough (a proportion of the rice base, prepared by mixing rice flour with tap water in a ratio of 1:1 (*w*/*v*, g/mL), was replaced with fermented sourdough at a proportion of 10, 20, or 30%) improved the specific volume as well as the fermentation properties of Indica and Japonica rice over 48 h compared to those of glutinous rice [22]. Research utilizing brown rice, which is more nutritious than white rice, has also been reported; brown rice sprouted for 24 h improves the physical properties of bread, and the addition of rice bran, which has a high soluble dietary fiber content, to ordinary rice flour increases the specific volume and improves the porosity of bread [46,47]. When brown rice is processed using a low-temperature impact mill, particle size reduction leads to further destruction of the starch crystalline structure and an increase in the damaged starch content. This alteration results in excessively sticky dough, imposing limitations on the dough expansion. Consequently, GF bread produced with medium-particle-size brown rice flour exhibits an elevated specific volume, diminished hardness, and uniform quality characteristics [48].

Cookies, cakes, and muffins constitute a substantial segment of bakery products. Remarkably, individuals with celiac disease are reported to consume more GF cookies and biscuits than GF bread [49]. Non-bread GF bakery items, such as cookies, cakes, and muffins, are predominantly manufactured utilizing GF flours such as rice, corn, and sorghum [11]. Nonetheless, several challenges persist in the development of GF bakery items, including technical barriers, nutritional quality concerns, and sensory attributes [5]. The quality of cakes and muffins is contingent upon factors such as the volume and crumb properties, which are intrinsically linked to the air bubbles incorporated into the dough during the mixing process [50]. Research aimed at enhancing texture and quality during the processing of cakes and muffins has primarily focused on the combination of starch, hydrocolloids, and proteins. The incorporation of resistant starch yielded no notable differences in the physical properties or sensory evaluation of rice flour cakes. Nonetheless, cakes containing xanthan gum, or a combination of xanthan and guar gum, exhibited a high porous area [51,52]. When 8 g/100 g of small-red cowpea protein was integrated into basmati rice, both the viscoelasticity and firmness of the muffin dough were augmented. Moreover, assessing the incorporation of various proteins into ordinary rice flour revealed that an increase in egg white protein and casein amplifies the specific volume of muffins [53,54]. The formation of a gluten network in cookies is notably constrained and predominantly diverges based on the type and attributes of the anticipated final product. Rice has been extensively utilized alongside other components to produce GF cookies marked by enhanced functionality and improved quality [55]. Cookies formulated by augmenting rice flour with alfalfa seeds, noted for their high protein and dietary fiber contents, exhibit a significant proportion of slowly digestible starch, and rice cookies with added annealed glutinous rice starch show the highest proportions of resistant starch and dietary fiber [56,57].

**Table 1 foods-12-04110-t001:** Rice-based GF bread and bakery products.

Product	GF Materials	Main Conclusions	References
Bread	Eleven commercial rice flours	Rice flour with low starch damage (<5 g/100 g) resulted in a high specific volume of bread.	[58]
	Rice flour (Fukumori Shitogi No. 2)	Bread treated with thermoase had a good crumb appearance and low staling rate and showed high specific volume.	[43]
	Polished japonica rice (Panjin, China)	Rice flour with a damaged starch content of 4–6%, which had a good gas retention capacity, resulted in a high specific volume, dense cell structure, and elastic bread texture.	[59]
	Short-grain rice flour Long-grain rice flour	The finest flours produced bread with a lower specific volume. Short-grain rice flour produced bread with higher specific volumes and lower firmness.	[60]
	Commercial certified brown rice cultivar (INIAP 15)	Flours obtained after 24 h of germination improved bread texture (more than 24 h of germination was not suitable).	[46]
	Japonica rice (Panjin, China)	Samples prepared by blending semidry-ground and dry-ground rice flour with particle sizes of 75–100 µm exhibited characteristics similar to those of wet-ground rice flour, as visualized via principal component analysis, and exhibited good bread volume and crumb texture.	[61]
	Wet-milled rice flour	Increased loaf volume and porosity of pre-gelatinized rice flour (50% water mixed with 1% rice flour at 80 °C for 2 min).	[44]
	Ten normal and glutinous rice cultivars	GF bread made using normal rice showed a higher specific volume and evener crumb cell distribution than those made using glutinous rice.	[62]
	Rice flour Four rice brans	The addition of rice brans, especially with high amounts of soluble dietary fiber, resulted in the higher specific volume, softer crumb firmness, and improved porosity of bread.	[47]
	Brown rice flour	GF bread with medium-sized brown rice flour prepared using a low-temperature impact mill showed a large specific volume, low hardness, and homogeneous gas cells.	[48]
	Commercial rice flour	Coarse rice particles (132–200 µm) with high dough hydration (90–110%) produced bread with a high specific volume.	[63]
	Commercial rice (Indica, Japonica, and waxy rice)	Adding 30% rice sourdough improved the specific volume of rice bread. Indica and Japonica rice bread with sourdough fermented for 48 h exhibited the highest cohesiveness and chewiness and the lowest hardness.	[22]
Cake	High-protein Frontière brown rice	High-protein brown rice showed the highest specific volume and was softer than the other rice flour cupcakes.	[64]
	Commercial rice flour	No significant difference in physical properties or results of sensory evaluation were observed for high-content resistant starch (RS)-added cakes. Highest RS (up to 20 g/100 g) added to cake is acceptable.	[51]
	Rice flour	A higher pore area fraction was observed in cakes containing a mixture of xanthan, guar gum, and xanthan gum. Cakes baked in an infrared microwave combination oven had high porosity.	[52]
Muffins	Basmati rice	Incorporation of (>8 g/100 g) cowpea protein isolates decreased flour paste viscosity and increased batter viscoelasticity, firmness, springiness, cohesiveness, and chewiness.	[53]
	Commercial rice flour	Casein and egg white protein increased the specific volume of muffins. Soy protein did not have any effect on the muffin texture, while pea protein-containing muffins were softer and springier.	[54]
Cookies	Commercial rice flour	Rice flour with alfalfa seed flour had higher protein and total dietary fiber contents. The cookies showed high slowly digestible starch properties.	[56]
	Rice flour, commercial native waxy rice starch	Cookies made from annealing-treated waxy rice starch had the highest dietary fiber and RS contents but the lowest rapidly digestible starch content.	[57]

GF, gluten-free.

### 3.2. Rice-Based GF Pasta and Noodles

Pasta is a tasty, easy-to-cook, and cost-effective dish, making it a staple food in various countries. Both pasta and noodles are simple to make, requiring only flour and water [29,65]. In several Asian countries, noodles serve as a staple food, constituting up to 50% of the total wheat consumption. Gluten plays a crucial role in conferring elasticity and chewiness to pasta, significantly influencing its cooking properties [66]. High-quality pasta/noodles are characterized by low cooking loss and stickiness [66].

Rice flour-based GF foods exhibit subpar cooking quality and sensory attributes due to the weak network formed by rice protein [67]. To address this issue, various treatments, such as hydrothermal treatment, enzyme treatment, and fermentation, along with the addition of enhancers, have been explored as potential gluten substitutes [68,69]. The absence of gluten hinders rice flour from forming a cohesive dough structure, leading to technical and quality challenges [29]. Consumers who prioritize nutritional content and have concerns about food additives prefer natural materials. Recent trends involve the incorporation of both plant- and animal-derived proteins, such as casein, egg white, and rice protein, into GF noodles and pasta. This addition aims to reduce the cooking loss and enhance texture [33,66,70]. Proteins, generally used as structural agents, strengthen the physical properties, mechanical strength, and stability of solid and semi-solid food items [67]. Despite the allergenic nature of some proteins proposed for GF food development, extensive research has been conducted on egg albumin, soy protein, and whey protein. Rice bran protein has emerged as a hypoallergenic and functionally superior alternative, serving as a high-quality protein source in a wide range of foods. Table 2 provides an overview of studies on rice-based GF pasta/noodle processing characteristics. Studies focusing on the enhancement of the pasta quality through protein addition have found that egg albumin significantly reduces the cooking loss and improves the firmness of pasta [66]. Additionally, integrating soy proteins into Indica-type rice spaghetti has been shown to reduce the stickiness, cooking loss, and cooking time [71]. The incorporation of 5–10% rice protein into high-amylose rice positively impacted its cooking properties. However, when this proportion was decreased to 2.5–5%, the sensory attributes closely resembled those of conventional wheat spaghetti [72]. Several studies have employed physical processing technologies for combinations of rice and other grains; these include a blend of rice and buckwheat for extruded pasta production, with adjustments made to processing conditions such as the optimal moisture and barrel temperature [73], and quality improvements in extrusion pasta crafted from parboiled Indica-type rice fortified with 25% amaranth powder and rice pasta enriched with 30% legumes [74,75]. In a study investigating the quality of rice noodles based on the milling technique used, it was observed that dry milling increases the damaged starch content, increasing the moisture solubility, which, in turn, increases the cooking loss [38].

### 3.3. Rice-Based GF Beer

Beer, the most popular alcoholic beverage globally, is primarily crafted from barley and wheat malt. However, for those with gluten intolerance, traditional barley beer poses safety concerns [17]. While enzymatic treatments can break down certain gluten proteins, they increase beer production costs and present technical challenges, potentially affecting the beer quality [76,77]. Brewers have a long history of using alternatives to barley as partial substitutes [78]. Many brewers incorporate adjuncts, such as rice and buckwheat, to introduce diverse flavors into their beers. In Korea, researchers have explored methodologies for sweetening rice beer and evaluated the quality characteristics of rice beers, each comprising 30% of different rice varieties [79,80].

Recently, researchers have increasingly focused on methods that use 100% alternative grains for GF beer production [81,82]. A challenge of using these alternative grains is the diminished extract yield in the wort and a deficiency in its nitrogen compounds [78], hindering fermentation and yeast growth [83]. When malted, rice produces α- and β-amylases that have activities surpassing those of the enzymes present in the raw form, showing potential for use in GF beer [84]. Malted rice contains a range of fermentable sugars and amino acids crucial for the brewing process. However, it has low soluble nitrogen and free amino nitrogen contents [85]. Table 3 presents studies that used rice as the main ingredient instead of barley malt for developing GF beer. Beer made from the malt of two Italian paddy rice varieties exhibited a pale hue, and the foam collapsed rapidly [86]. To address this issue, a specific thermal method was employed to caramelize the rice malt, enhancing its suitability for brewing [87]. The top-fermented beer prepared using rice malt possessed a high alcohol content and was devoid of any undesirable flavor [81]. Notably, rice malt was reported to produce more maltose than glucose [85].

## 4. Additives and Processing Technologies for Rice-Based GF Foods

### 4.1. Ingredients as Nutritional Enhancers or Gluten Substitutes

Rice flour stands out as the most used GF flour in GF food products. However, to optimize the dough properties and the overall quality of GF products, formulations often extend beyond rice flour to include flours, starches, and proteins sourced from other grains, pseudocereals, pulses, and plant-based materials. While factors such as the milling method, flour granularity, and treatment processes may have some influence, the compositions and formulations of these complex GF flours predominantly shape the physical properties and sensory profiles of the resulting products [5]. As the demand for GF products has surged, so has the research into diverse types of GF grains, considering their nutrition, digestibility, and usability. GF grains have been typically used in traditional cuisines within their native regions for centuries. Today, they are being increasingly applied in the development of GF foods for individuals with celiac disease and gluten sensitivity. Such GF grains may contain nutrients superior to those in gluten grains, such as wheat. However, the absence of gluten in these grains compromises their processing capabilities, which can adversely impact the quality of the final product [87]. Thus, solely relying on a blend of GF grains has certain limitations, and the use of functional ingredients, such as starches and hydrocolloids, is a prevalent practice [88].

In the food industry, hydrocolloids, which are long-chain polymers formed by polysaccharides and proteins, serve as thickeners to enhance the quality and shelf lives of a variety of common foods, including soups, sauces, and jams [16,89]. These hydrocolloids can bind moisture and thus play an essential role in replenishing moisture in food during the cooking process. They have been noted to improve the cohesiveness and consistency of the quality of pasta [90]. Starch is another common additive, consisting of amylose and amylopectin, the proportions of which differ based on the source grain. The quality of starch is instrumental in determining the rheological properties of dough and, by extension, the final quality of foods, including bread [91]. Although starch, similar to gluten, can trap carbon dioxide during gelatinization, supporting dough rheology, it cannot replicate the expansion (rising capacity) and viscoelasticity provided by gluten [91]. The use of proteins in the production of GF products has emerged as a promising approach. These proteins are either native to GF flours, such as rice and soy, or introduced in diverse forms, such as concentrates and isolates [92]. Incorporating non-gluten proteins into GF products positively influences not only their structure, texture, and sensory profile but also their nutritional quality, mitigating potential amino acid deficiencies [93,94].

Another additive successfully used in GF foods is enzymes. These are utilized for their unique ability to modify the protein functionality and promote protein cross-linking [95,96].

Substituting rice flour with 30% quinoa flour enhanced its protein and mineral contents, decreasing the specific volume of the muffin while increasing the density, firmness, chewiness, and elasticity [97]. Studies evaluating the incorporation of various hydrocolloids at different concentrations during the production of rice cakes revealed that 1.5% balangu seeds, 1% xanthan, and 1% basil are effective [26,98]. The addition of rice bran protein concentrate (2% *w*/*w*) to rice flour improves the specific volume, crumb porosity, and sensory profiles of the final product, and the quality is reportedly better than that obtained when using egg albumin [99]. Moreover, when rice flour was supplemented with egg whites, whey protein, peas, and rice protein in the range of 15–45% during the production of cakes, the egg whites and whey proteins increased the firmness, cohesiveness, and springiness of the cakes; in contrast, pea and rice proteins reduced the firmness and cohesiveness [100]. Notably, muffins made from rice flour containing casein and egg whites showed an increase in specific volume, while the addition of soy protein isolate had no effect on their texture, and pea protein isolate made them softer [54].

### 4.2. Processing Technologies

The most common approach to improving the quality of GF products is to modify the macro-molecular structure of the starch. As the primary component of GF ingredients, starch plays a pivotal role in GF food production. GF food technology primarily relies on dough heating and cooling processes that leverage two phenomena: pre-starch gelatinization and its subsequent retrogradation [16]. Table 4 details a recent study on the application of rice-based GF food applying various processing methods and raw materials. One or two processing methods suitable for each product and GF grain materials for rice replacement, as well as hydrocolloids and proteins as gluten substitutes, were used. GF pasta produced via the extrusion process displayed improved quality, characterized by enhanced firmness and texture after cooking and reduced cooking loss. Hot-air-dried pasta showed increased firmness and protein solubility, and a decrease in cooking loss and viscoelasticity [101,102]. Physical treatments, including annealing and hot-water treatment, have often been applied in rice noodle production [103]. Annealing is a heat treatment technique that involves the low-temperature gelatinization of rice at moisture levels above 40% in the temperature range of 50–60 °C. Hydrothermal treatment of rice involves high-temperature processing at 100–120 °C with a relatively small amount of moisture. Such hydrothermal treatments inhibit starch granule swelling, delay gelatinization, enhance the stability of starch paste, and improve the texture and cooking properties of rice noodles [104]. Moreover, heat-treated flour used in GF bread production increases the dough viscosity, resistance, and elasticity, as well as the bread volume. However, a high-pressure treatment at 600 MPa reduced the specific volume and loaf volume, decreasing the bread quality [105,106]. In recent years, several technical processes have been applied to adjust the characteristics of GF dough and enhance the baking suitability. High-pressure treatment in the range of 100–1000 mPa lowered the gelatinization temperature of starch, changing the protein characteristics, including cross-linking, and modifying the viscoelasticity of the dough [107]. Techniques such as ultrasonication and micromilling were used to reduce the flour particle size but had no positive effect on the bread volume or porosity [108]. Pre-gelatinization through thermal treatments, such as the microwave heating of rice flour with 20–30% moisture, increased the dough flexibility, gas retention capacity, and dough viscosity [109]. The pressure reduction and high-temperature treatment (100–165 °C) of a rice and soy flour mixture set conditions similar to those of wheat bread [110]. Although infrared treatment permits less energy penetration, it ultimately results in a superior sensory evaluation [111].

In beer production, using an alternative carbohydrate source instead of barley malt can reduce the gluten content in the beer. However, a simple replacement with GF grains alone cannot achieve the desired effect because it is essential to activate the enzymes that can decompose starch into sugar during the malting process of GF grains. This highlights the compelling need to establish relevant processing technology. Malt, a primary ingredient in beer production, is germinated barley and can produce a significant amount of α–amylase, β-amylase, protease, and α–glucosidase. Hence, the germination process enhances the nutritional value and usability of the ingredients [17]. The malting conditions for rice include steeping, germination, and kilning, which have been investigated over the past years [84]. Optimal conditions have been selected through experimental evaluation with the following conditions: steeping rice for 7–48 h (range: 15–28 °C) and allowing germination to persist for 4–8 days (range: 14–30 °C) and kilning to continue for 8–24 h (range: 30–70 °C) [87,112]. However, even when malted, the activity of α–amylase and β-amylase in GF grains, including rice, is lower than that in barley. This discrepancy poses a considerable challenge for GF beer development. Therefore, brewing procedures must be adapted to cater to the unique attributes of each GF substitute grain [17]. Sensory assessment is of paramount importance in the development of GF rice beer. Research utilizing rice malt has indicated that, during top fermentation, the resulting beer often possesses a weak body, off-flavors, fruity undertones, and a solvent-like taste. Furthermore, the beer tends to be strongly bitter with minimal foam stability; however, the evaluation results were generally accepted as satisfactory. Thus, the quality characteristics can differ based on the production methods and technologies employed [81,86,113]. Therefore, GF food development requires the continuous evaluation of the characteristics by applying new and diverse technologies, with a focus on processing techniques more suited for rice starch.

**Table 4 foods-12-04110-t004:** Different ingredients and processing methods used for rice-based GF products.

Processing Methods	GF Materials/Additives	Major Conclusion	References
Baking	60% high-protein Frontière brown rice flour, 30.71% tapioca starch, 9.29% potato starch	Addition of starch to the cupcake formula decreased the hardness and increased the specific volume of the cupcakes.	[114]
Baking	Rice flours, tamarind gum, modified tamarind gum (MTG; thermoresponsive xyloglucan), and xanthan gum	The bread with MTG added was soft, moist, and preferred over those with other additives. The addition of 0.5–0.75% polysaccharides inhibited the hardening and aging of the bread with MTG added.	[115]
Baking	Rice flour, rice protein, hydroxypropyl methylcellulose (HPMC)	Amounts of 3% rice protein (RP) and 2% HPMC improved the quality of GF bread. RP provided a moist, springy, and resilient crumb matrix of the bread.	[116]
Baking	Broken white rice grains, carboxymethyl cellulose	Baking temperature of 185 °C, baking time of 22 min, and 0.8% concentration of carboxymethyl cellulose are desirable conditions for the development of gluten-free, low-glycemic-index cookies from rice flour.	[117]
Baking, hydrolysis (enzyme)	Rice flours, hydrolyzed peanut protein	The incorporation of 5% hydrolyzed protein had a positive impact on the specific volume and relative elasticity of gluten-free bread.	[118]
Baking, extrusion	Extrusion-cooked red rice flour	An increase in extrusion temperature increased the attractive red color of samples. The acceptance of the sensory properties was improved with the incorporation of extrusion-cooked flour.	[119]
Extrusion	Rice, buckwheat (partial replacement: 0, 15, 30, 45, 60 g/100 g)	The quality of buckwheat-added rice noodles (BRNs) improved, dietary fiber increased, and the cooking loss and broken rate of 30 g/100 g BRNs were the lowest.	[120]
Extrusion	Red and black rice	The optimum conditions, which are feed moistures of 15.5% and 16.0% and temperatures of 159 and 150 °C for black and red rice extrudates, respectively, resulted in cereal breakfast balls with optimal water solubility, volume, and texture and good color.	[121]
Steaming	Rice flours, *Apios americana* tubers (Apios)	Addition of Apios powder (a high level of β-amylase activity) to gluten-free bread improved bread texture and delayed staling.	[122]
Ultrasound, microwave, hydrothermal	Semidry-milled rice flour	Ultrasound treatment improved the bread quality (the specific volume increased by 15.6%, and the hardness decreased by 17.6%).	[123]

GF, gluten-free.

## 5. Conclusions

We conducted a comprehensive literature review on the use of rice in GF food research and technological advancements. Rice, the most widely consumed staple food globally, serves as a versatile GF substitute. However, outcomes vary based on factors such as the grain processing techniques, including particle size and milling methods, and inherent nutritional properties, such as in brown rice with its high protein content and amylose levels. To optimize the key attributes in processed products, such as dough leavening in bread and the physical and cooking properties of noodles and pasta, research has focused on combining different GF grains and incorporating additives that mimic the gluten function. Additionally, various processing technologies, such as starch preprocessing and extrusion puffing processes, have been employed to boost the quality of rice-based GF products. Today, a variety of products, including bread, noodles, and beer, use rice as a partial replacement for barley or wheat, and a noticeable proportion of consumers are showing a preference for products containing rice as a substitute. However, health-conscious consumers are not only concerned about gluten but also other allergens, and products that excel both in sensory appeal and functionally are in high demand. Thus, meeting these expectations through innovative GF product development is of paramount importance.

## Figures and Tables

**Table 2 foods-12-04110-t002:** Studies on rice-based GF pasta/noodle processing characteristics.

Product	GF Materials	Main Conclusions	References
Pasta	Rice flour	Pasta with 9% egg albumin had the lowest cooking loss, with improved firmness.	[65]
	Rice–buckwheat flour mixture	To produce good-quality pasta (low cooking loss and stickiness and good hardness and firmness), optimal conditions (30% moisture content, 120 °C barrel temperature, and 80 rpm screw speed) were established.	[73]
	Parboiled milled rice (Indica cultivar)	Extruded cooking pasta with 25% amaranth flour exhibited increased firmness and decreased protein solubility.	[74]
	Rice flour	Precooked spaghetti with legume-containing rice flour (up to 30 g/100 g) showed low cooking loss (<6%) and acceptable sensory score and was nutritionally valuable.	[75]
	Two Indica cultivars	Rice spaghetti with 5% soy protein isolate showed increased firmness and decreased stickiness, cooking time, and cooking loss.	[71]
	High-amylose rice cultivar	Rice spaghetti containing 5–10% rice protein showed decreased cooking time and loss. Sensory evaluation was comparable to wheat spaghetti when 2.5–5% RP was added.	[72]
Noodles	Commercial polished Japonica rice	Dry-milled rice flour showed high damaged starch content, increasing the water solubility of rice noodles, and resulting in increased cooking loss.	[38]
	Commercial rice flours	Rice noodles with mushroom β-glucan-rich fragments exhibited reduced cooking loss and enhanced extensibility and firmness.	[29]

GF, gluten-free.

**Table 3 foods-12-04110-t003:** Studies on rice-based GF beer processing characteristics.

Product	GF Materials	Main Conclusions	References
Beer	Two Italian paddy rice varieties	The color of the rice malt beer was pale, and the foam collapsed rapidly. The sensory profile was flat compared to barley malt beer and needed to be improved.	[86]
	Two Italian paddy rice varieties	Top-fermented beer had high alcohol content and low ester content, without off-flavors.	[81]
	Italian paddy rice (japonica)	The color and flavor of roasted special rice malts (caramelized and dark) were enhanced and showed high polyphenol content.	[87]
	Rice and buckwheat	Rice malts consistently produced more maltose than glucose. Buckwheat malts produced more glucose and free nitrogen than rice malts.	[85]

GF, gluten-free.

## Data Availability

The data presented in this study are available upon request from the corresponding author.

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
