# Peer review of "Rice-Based Gluten-Free Foods and Technologies: A Review"

_foods, 2023, doi:10.3390/foods12224110_

Round 1

Reviewer 1 Report

Comments and Suggestions for Authors

This is a review paper about the use of rice in gluten-free foods.  The paper is well written and provides good information to the reader. It will be good to add a few more references comparing rice to other grains used in gluten-free foods. Also See below for some minor edits.  Red is remove, yellow is add.   Two issues to address:

Lines 8 and 74 contradict each other as to which grain is the most consumed world wide.  Also, line 138 mentions "grain hydrocolloids". I am not aware of any hydrocolloids produce at an industrial level from grains

8 Rice, the most widely consumed staple worldwide

46 reach USD 8.3 billion by 2025 in the U.S.

55 Numerous review articles have scrutinized discussed

74 Rice, following corn and wheat, is among the most extensively consumed grains worldwide

109 augmented wheat flour consumption

138 grain hydrocolloids

Comments on the Quality of English Language

Good command of the English language. Article is easy to read and follow

Author Response

This is a review paper about the use of rice in gluten-free foods.  The paper is well written and provides good information to the reader. It will be good to add a few more references comparing rice to other grains used in gluten-free foods. Also See below for some minor edits.  Red is remove, yellow is add.  Two issues to address:

We appreciate your careful review of our manuscript and have revised the manuscript according to the reviewers’ comments.

 According to your opinion, the difference in the component characteristics of GF grain and rice and considerations for product development have been presented (Line 55-61). In addition, since this study is a rice-based GF study, we describe the results of mixing GF grains as a substitute for rice in Table 4. We have responded to each of the comments as follows.

Lines 8 and 74 contradict each other as to which grain is the most consumed world wide.  Also, line 138 mentions "grain hydrocolloids". I am not aware of any hydrocolloids produce at an industrial level from grains

8 Rice, the most widely consumed staple worldwide

→ Thank you for pointing this out. We have added “one of” .

46 reach USD 8.3 billion by 2025 in the U.S.

→ Thank you for your comment. We updated this part with the latest statistics. It has been revised to the contents of the global market as of 2022-2032, on lines 47-78.

55 Numerous review articles have scrutinized discussed

→ Thank you for your suggestion. We have revised it accordingly (Line 62)

74 Rice, following corn and wheat, is among the most extensively consumed grains worldwide

→ Some text has been added on line 8.

109 augmented wheat flour consumption

→ We have revised the sentences on lines 116-117.

138 grain hydrocolloids

→ Thank you for pointing this out. We have deleted it (Line 146)

Comments on the Quality of English Language

Good command of the English language. Article is easy to read and follow

→ Thank you for your opinion. We sought professional help for English language editing.

Reviewer 2 Report

Comments and Suggestions for Authors

The authors summarize the recent reports about the researches of rice-based gluten-free foods. This paper is well written and easy to follow. It should be of interest to the researchers in food science. However, I have a comment that the authors might consider.

Comment: A number of the references are incorrect.
I think that reference [5], [7], [8], [9], [10], [11], [12], [41], and [47] are not appropriate. For example, the author quoted reference [9] in lines 45-46 as a prediction of gluten-free consumption. However, reference [9] indicates gluten-free labeling of foods.
Reference [17] and [36] is missing in the manuscript.
Please check reference list and manuscript thoroughly.

Author Response

The authors summarize the recent reports about the researches of rice-based gluten-free foods. This paper is well written and easy to follow. It should be of interest to the researchers in food science. However, I have a comment that the authors might consider.

Comment: A number of the references are incorrect.
I think that reference [5], [7], [8], [9], [10], [11], [12], [41], and [47] are not appropriate. For example, the author quoted reference [9] in lines 45-46 as a prediction of gluten-free consumption. However, reference [9] indicates gluten-free labeling of foods.
Reference [17] and [36] is missing in the manuscript.
Please check reference list and manuscript thoroughly.

We appreciate your review of our manuscript. We totally agree with you and apologize for the confusion caused by the errors in the references. We have added some references and reviewed them again; furthermore, we have revised the reference numbers.

Reviewer 3 Report

Comments and Suggestions for Authors

This manuscript reviewed rice-based GF foods. Based on my search on WOS, similar review article focusing on rice is not available. In this regard, this concise review is valuable. However, I feel the manuscript is a little bit hard to read, possibly due to the article that can be improved. My major concerns are: 

1. I do not understand the difference in Table 1 and Table2. In table 1, there is also some information about processing technologies. Thus, I feel that these two tables can be combined or re-formatted so that one table is concerned with effects of ingredients and another is concerned with processing effect. 

2. It seems that major portion of cited studies are from 5 years ago. Authors are suggested to double-check if they missed some recent papers as rice-based GF product is a popular topic.

3. There is no table or figure showing information in Section 4. But section 4 is an important part in this manuscript. Generally, it's felt that the connection between Section 3 and Section 4, Table 1 and Table 2 are not clear. 

Author Response

Comments and Suggestions for Authors

This manuscript reviewed rice-based GF foods. Based on my search on WOS, similar review article focusing on rice is not available. In this regard, this concise review is valuable. However, I feel the manuscript is a little bit hard to read, possibly due to the article that can be improved. My major concerns are: 

  1. I do not understand the difference in Table 1 and Table2. In table 1, there is also some information about processing technologies. Thus, I feel that these two tables can be combined or re-formatted so that one table is concerned with effects of ingredients and another is concerned with processing effect. 

We appreciate your review of our manuscript. We agree with you in part. Before we wrote this review manuscript, we considered and carefully reviewed what classification criteria to write and how to organize the paper.

First, for your understanding, I am briefly explaining this review paper (the overall schematic diagram of this review is shown in Figure 1). We found rice-based gluten-free application studies, presented the study contents of gluten-free products in section 3, and divided them into Tables 1, 2, and 3. In general, research papers on gluten-free products can be broadly divided into bakery products, noodles, and beverages, and bakery products are classified into bread and other cakes, muffins, and cookies (or biscuits) and are being studied in detail around the world. Noodles are divided into pasta and noodles, and pasta is studied in various forms such as spaghetti and penne, which are mainly prepared by the extrusion process. Among gluten-free drinks, the main subject studied is beer containing alcohol. The main technology used to prepare beer is malting.

Therefore, we found relatively recent papers and organized them by classifying them on a product basis. Thus, as you mentioned, Tables 1, 2, and 3 include the pre-treatment and processing conditions of rice raw materials, mixing of other ingredients, and the use of gluten replacements in the study of gluten-free products. Table 1 lists rice-based GF breads and bakeries, Table 2 rice-based GF noodles, and Table 3 rice-based GF beers, and they are discussed in Sections 3.1, 3.2, and 3.3, respectively. The latest rice-based GF product developments are summarized to be included in Tables 1, 2, and 3, and these research papers also include information on ingredients and processing technologies. I hope this explanation will help you better understand the review.

  1. It seems that major portion of cited studies are from 5 years ago. Authors are suggested to double-check if they missed some recent papers as rice-based GF product is a popular topic.

→ According to your comment, we recently added a total of 9 references related to rice-based GF research.

  1. There is no table or figure showing information in Section 4. But section 4 is an important part in this manuscript. Generally, it's felt that the connection between Section 3 and Section 4, Table 1 and Table 2 are not clear. 

→ Thank you for your opinion. As we answered the first comment, Tables 1 and 2 are covered in Section 3 and are clearly distinguished because they are for different products (bakery products including bread, pasta and noodle products). Please review it again. In addition, following your careful comments, we have added Table 4 to the latest rice-based GF-related studies corresponding to Section 4.

Round 2

Reviewer 2 Report

Comments and Suggestions for Authors

The authors have responded appropriately to my comments in a short period of time.
I have no further comments.